# EGFR Mutation in Nasopharyngeal Carcinoma

**Evren Uzun * and Suna Erkilic**

Department of Pathology, Medical Faculty, Gaziantep University, Gaziantep 27410, Turkey
* Correspondence: drevrenuzun@gmail.com; Tel.: +90-3423606060

**Abstract:** Nasopharyngeal carcinoma is a malignant tumor of the nasopharynx. However, while radiotherapy is the primary choice of treatment, the treatment may fail due to distant metastasis in most patients at an advanced stage. Treatment agents against some mutations have led to the development of personalized treatment regimens. EGFR is one of the most studied molecules and has played a role in the development of a large number of cancer types. We aimed to demonstrate the EGFR mutation status in nasopharyngeal carcinomas. Twenty-six nasopharyngeal carcinomas were included in the study. EGFR mutation analysis was applied to the cases by the real-time PCR method. The results were evaluated statistically. No EGFR mutation was detected in any of the cases. Although EGFR expression is frequently shown in nasopharyngeal carcinomas immunohistochemically, the same positivity was not shown in genetic analysis. This result shows that the use of anti-EGFR agents in nasopharyngeal carcinoma treatment will not be effective.

**Keywords:** nasopharynx; squamous; carcinoma; EGFR; mutation; nasopharyngeal carcinoma





## 1. Introduction

Nasopharyngeal carcinoma (NPC) is the primary tumor of the nasopharyngeal epithelium. NPC is an endemic disease that is common in China, Southeast Asia, and North Africa [1].Etiological considerations include the Epstein–Barr virus (EBV), genetic, and environmental factors [2]. Based on the World Health Organization (WHO) histopathological classification, these tumors are divided into three subgroups. These are keratinizing, non-keratinizing, and basaloid squamous cell carcinomas [3].Keratinizing NPCs are rare in non-endemic areas, whereas non-keratinizing NPCs are more common in endemic areas that are closely associated with the Epstein–Barr virus [4].

In early-stage carcinomas, radiotherapy (RT) is the primary treatment choice. The five-year survival rate is over 90% in these tumors. However, 70% of patients have locally advanced disease at the time of diagnosis. Combined RT and conventional chemotherapy (CT) is the recommended treatment for these patients. In 30% of patients receiving combined therapy, treatment fails due to metastatic disease [5,6]. Because of this inefficacy, new treatment alternatives are needed. With the development of technology and molecular medicine, the number of targeted therapy agents used in cancer therapy has increased. Specifying a specific target for the development of therapies is critical for improving patient survival and prognosis. Therefore, understanding cancer biology is the most critical step in determining targeted treatment options [7,8]. Although there have been some chromosomal abnormalities and amplification of specific oncogenes detected in NPC, information on oncogenic mutations in NPC is limited. EGFR mutation is a molecule frequently studied in NPC; additionally, studies are frequently immunohistochemical. Molecular analyses are limited and contain controversial results. This inconsistency makes it ambiguous whether EGFR is a potential target in NPC patients [9]. In this research, we aimed to show the EGFR mutation status of nonkeratinizing nasopharyngeal carcinomas.

## 2. Materials and Methods

Patient selection: Twenty-six nasopharyngeal carcinoma cases diagnosed between 2015 and 2017 were included in this study. Age, sex, tumor size, tumor location, additional disease, diagnosis of previous biopsies, EBV (EBER in situ hybridization) status and lymph node metastasis status were investigated. Tumor histopathology was reviewed using H&E-stained slides in accordance with WHO nasopharyngeal tumor classification [3].

DNA extraction: Hematoxylin-eosin (H&E)-stained sections with a tumor cell ratio of 50% or more were selected. An average of 5–10 sections of 4–6 micron thickness were taken from the paraffin blocks of these sections and transferred to 1.5 mL volume Eppendorf tubes. Eppendorf tubes were kept at room temperature until DNA extraction. The DNA extraction was performed using the Cobas DNA sample preparation kit (Roche, Basel, Switzerland). The isolated DNA quality and quantity were measured by the Nanodrop ND-2000 spectrophotometer (Thermo Scientific, Niederelbert, Germany). The amount of DNA sufficient for EGFR mutation analysis was accepted as five ng/dl.

*Molecular Analysis*

Samples prepared with a DNA sample preparation kit were analyzed with thereal-time PCR method (Cobas 480, Roche, Basel, Switzerland). G719X (G719A, G719C, and G719S) mutation in exon 18, deletions and complex mutations in exon 19, S768I, T790M mutations, and insertions in exon 20, L858R, and L861Q mutations in exon 21 were examined (Cobas EGFR mutation analysis test).

## 3. Results

Patient characteristics: Twenty-six NPC cases were included in the study. Three of these patients were female, and twenty-three were male. The mean age of the patients was 46 (range 14 to 74). According to WHO, all cases were classified as nonkeratinizing squamous cell carcinoma. Most of the patients presented with complaints of neck swelling, while a few patients presented with hearing loss. Ten cases had cervical lymph node metastasis at diagnosis. All cases were EBV positive with EBER in situ hybridization.

Molecular results: All cases were analyzed by the real-time PCR method for potential mutations in EGFR exons 18,19,20, and 21.No EGFR mutation was detected in any of the cases.

## 4. Discussion

NPC is a malignant tumor of the nasopharyngeal epithelium and is common in Southeast Asia, China, and North Africa. Nitrosamines, consumption of salted fish, EBV infection, and many other etiological factors have been put forward to explain the geographic distribution. NPC is most commonly seen between 40 and 60 years of age and has male dominance. The presenting complaints are swelling in the throat, symptoms related to nasal obstruction, serous otitis, and tinnitus [10]. According to WHO classification, NPC is classified into keratinizing, nonkeratinizing, and basaloid subtypes [3]. Keratinizing NPCs have the classical features of squamous cell carcinoma, such as keratinization, keratin pearls, and intercellular bridging. Nonkeratinizing NPCs are characterized by solid layers and irregular islands, and are divided into differentiated and undifferentiated subtypes. The differentiated subtype is similar to transitional cell carcinoma of the bladder and includes a paving-stone appearance and cellular stratification. The undifferentiated NPC consists of large cells with prominent nucleoli and round-oval vesicular nuclei in a syncytial pattern. Tumor cells are usually associated with lymphoid stroma. Non-keratinizing NPC is more associated with EBV infection. It is more radiosensitive and more closely associated with lymph node metastasis and distant metastasis [11].

Most NPC patients are at an advanced stage at the time of diagnosis. This situation is associated with short survival and increased recurrence and metastasis despite combined RT and CT treatment. As a result, alternative therapies except RT/KT are needed to improve the survival and prognosis of NPC patients regardless of their stage [5,6]. Recently,

the identification of mutations that contribute to the development of solid tumors has resulted in an increase in the number of targeted therapeutic agents developed to combat these mutations. In addition to being less cytotoxic than conventional chemotherapy, these agents have improved the prognosis and survival of patients [12,13]. Therefore, oncogenic mutation analysis studies to determine alternative treatment targets in NPC patients are essential. These studies generally examined NPCs in terms of common mutations seen in other solid organ tumors. One of the most studied target mutations is the EGFR mutation in all cancer biology. In the last two decades, epidemiological and experimental studies have shown that abnormal EGFR expression and signaling have an essential role in the development of cancer. The presence of EGFR mutations has created a good treatment option, especially in non-small cell lung carcinomas [12]. Activation of EGFR stimulates cell proliferation and angiogenesis.

Additionally, this activation induces invasion and metastasis ability, making cells protected from apoptosis that results in resistance to chemoradiotherapy [14,15]. EGFR mutations are grouped into three titles: mutations that cause changes in the extracellular domain, mutations that cause changes in the intracellular domain, and mutations that cause changes in the intracellular tyrosine kinase domain. The main interest is in the last group, where EGFR mutations are grouped into four regions: exon 18, 19, 20, and 21. More than ten agents targeting EGFR are used in the treatment of various cancers. The most interesting EGFR tyrosine kinase inhibitors are gefitinib and erlotinib. These two oral active EGFR TKIs are effective in non-small cell lung carcinomas. Anti-EGFR monoclonal antibodies bind to the extracellular domain of EGFR inhibit activation of pathways and exhibit antineoplastic behavior [16].

In the literature, EGFR overexpression is reported in about 80% of NPCs immuno-histochemically [12,17,18]. EGFR-overexpressed tumors have been shown to have shorter survival and worse prognosis and were higher in nonkeratinizing NPCs [19]. This suggests that EGFR is a potential target for NPC treatment. However, EGFR mutation could not be shown at the same frequency in genetic analysis.

In studies that analyzed 60 NPCs in Moroccan patients (58 undifferentiated, one differentiated nonkeratinizing, and one keratinizing NPC) and four NPC cell lines, there was no EGFR mutation detected [20,21]. In other publications, the EGFR mutation frequency in nonkeratinizing undifferentiated NPCs was reported at 4.3% and 1%. These mutations were the T790M mutation in exon 20 and the E709A mutation in exon 20, respectively [22,23].

In a comprehensive study; of the 160 NPC patients, EGFR mutations were detected in five tumors (2 N771_P772>SVDNR, 1 T790M, 1 H773_V774insNPH, 1 R108K). In another study, despite the presence of silent mutations in 102 NPC patients, a potential target mutation was not detected [24,25]. In EGFR-positive tumors, there was no association between mutations and clinicopathological characteristics such as age, gender, histological subtype, EBV infection, and clinical stage [22–24]. In the present study, we analyzed EGFR mutations in 26 non-keratinized NPCs, and no mutation was detected.

Considering the literature in terms of non-EGFR mutations seen in NPC, CDK4, KIT, PDGFRA mutations, and less frequently KRAS, BRAF, MET, FGFR3, AKT1, and PIK3CA mutations, were detected. Additionally, there was no relationship between these mutations and clinicopathological characteristics [22,23,25].

EGFR mutations play a role in the development of some organ malignancies, particularly non-small cell lung carcinoma, and anti-EGFR agents are effective in this case. The lack of detection of EGFR mutation in analyses studied on NPC, including our study, supports the idea that EGFR mutation is rare in non-lung solid human tumors. In studies with cervix, colorectal, gastric, breast, acute leukemia, glioblastoma and hepatocellular carcinomas, EGFR mutation was found in one breast, one colorectal, and one glioblastoma. All these results show that EGFR targeting TKI therapies (erlotinib and gefitinib) are not useful in NPC and extrapulmonary malignancy treatment [26]. Although NPC is a radiosensitive tumor, alternative treatment methods are needed, especially in patients who have failed treatment due to metastasis and recurrence. Nowadays, with the development

of technology, the scope of genetic analysis has expanded, and many mutations can be used as targets in cancer treatment. The agents developed for the targets detected in tumors such as colorectal carcinomas, breast carcinomas, melanoma, and primarily non-small cell lung carcinomas are used as alternatives to conventional and radiotherapy. It has been reported that mutations known to play a role in the development of carcinoma are not frequently shown in NPC studies. One of them is the EGFR mutation, and in some studies, it was found in a few cases, and in most studies, it was not seen at all. In NPCs we did not detect any EGFR mutation. These studies showed that the incidence of oncogenic potential target mutations in NPC is less common in other organ tumors. Further and comprehensive studies are needed to determine the efficacy of therapies that can be developed for rare mutations or to identify new potential targets.

**Author Contributions:** Conceptualization, S.E. and E.U.; methodology, S.E.; software, E.U.; validation, S.E. and E.U.; formal analysis, E.U.; investigation, E.U.; resources, S.E.; data curation, E.U.; writing—original draft preparation, E.U.; writing—review and editing, E.U.; visualization, E.U.; supervision, S.E. All authors have read and agreed to the published version of the manuscript.

**Funding:** This research received no external funding.

**Institutional Review Board Statement:** Ethical review and approval were waived for this study, due to the totally retrospective nature of the research.

**Informed Consent Statement:** Patient consent was waived due to the totally retrospective nature of the research.

**Data Availability Statement:** Not applicable.

**Conflicts of Interest:** The authors declare no conflict of interest.

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
