# Peer review of "EGFR Mutation in Nasopharyngeal Carcinoma"

_jmp, doi:10.3390/jmp3040017_

Round 1
Reviewer 1 Report
A negative finding of molecular alterations of EGFR in nasopharyngeal carcinoma cases, similar to what has already been reported.
The current 2022 WHO listed non-keratinizing squamous cell carcinoma as the category, and this should be used instead of differentiated and undifferentiated.
The results should include EBV status (EBER reaction) -- to help with EBV vs. HPV or other viral etiologies.
There are several syntax, grammar and English language errors that need to be corrected by employing a native English language review.
Author Response
RESPONSE TO REVIEWER 1
1- In the literature, about 20 studies have performed EGFR expression analysis immunohistochemically in nasopharyngeal carcinomas. On the other hand, there have been fewer than 10 studies that have genetically studied EGFR mutations. Most of these genetic studies involve simultaneous mutational analysis of many tumor types, including nasopharyngeal carcinomas, and some have reported low numbers of EGFR mutant tumors. Although our study is one of three studies directly targeting EGFR mutation in nasopharyngeal carcinomas, it is the only study studied in the nonkeratinized subgroup and analyzed by the Real-time PCR method. Additionally; although high positive results were obtained in EGFR immunohistochemical expression analyses, lower than expected and contradictory results were obtained in anti-EGFR treatment studies performed in nasopharyngeal carcinomas. In addition, most of the anti-EGFR treatment studies included not only type 2 carcinomas but also all groups and did not include EGFR immunohistochemical expression information for the cases.
2- Differentiated, dedifferentiated subtype definitions in the study were removed.
3- EBV results of the cases have been added to the findings section.
4- It was reviewed by a native English speaker and necessary corrections were made.
Reviewer 2 Report
The work presents the already known findings. No novelty items, small group of patients.
Author Response
RESPONSE TO REVIEWER 2
In the literature, about 20 studies have performed EGFR expression analysis immunohistochemically in nasopharyngeal carcinomas. On the other hand, there have been fewer than 10 studies that have genetically studied EGFR mutations. Most of these genetic studies involve simultaneous mutational analysis of many tumor types, including nasopharyngeal carcinomas, and some have reported low numbers of EGFR mutant tumors. Although our study is one of three studies directly targeting EGFR mutation in nasopharyngeal carcinomas, it is the only study studied in the nonkeratinized subgroup and analyzed by the Real-time PCR method. Additionally; although high positive results were obtained in EGFR immunohistochemical expression analyses, lower than expected and contradictory results were obtained in anti-EGFR treatment studies performed in nasopharyngeal carcinomas. In addition, most of the anti-EGFR treatment studies included not only type 2 carcinomas but also all groups and did not include EGFR immunohistochemical expression information for the cases.
Reviewer 3 Report
The Authors reports on a retrospective observational study which tested the mutational status of the EGF receptor in nasopharyngeal cancer patients. I had a careful look at the manuscript, and I am a bit skeptical about the overall value of the present study. Most of the efficacy of anti-EGFR therapy in HN cancer is derived by overexpression of EGFR in squamous cell histology. In the present study, NPC WHO type II is analyzed. I would expect EGFR involvement in pathogenesis of NPC type I. Moreover, I am not sure EGFR mutational status, rather than overexpression, is of interest.
Author Response
RESPONSE TO REVIEWER 3
In the literature, about 20 studies have performed EGFR expression analysis immunohistochemically in nasopharyngeal carcinomas. On the other hand, there have been fewer than 10 studies that have genetically studied EGFR mutations. Most of these genetic studies involve simultaneous mutational analysis of many tumor types, including nasopharyngeal carcinomas, and some have reported low numbers of EGFR mutant tumors. Although our study is one of three studies directly targeting EGFR mutation in nasopharyngeal carcinomas, it is the only study studied in the nonkeratinized subgroup and analyzed by the Real-time PCR method. Additionally; although high positive results were obtained in EGFR immunohistochemical expression analyses, lower than expected and contradictory results were obtained in anti-EGFR treatment studies performed in nasopharyngeal carcinomas. In addition, most of the anti-EGFR treatment studies included not only type 2 carcinomas but also all groups and did not include EGFR immunohistochemical expression information for the cases.
Round 2
Reviewer 2 Report
The work is still very laconic: the detailed characteristics of the examined patients (usually presented in the table) is not presented, no description of the test with which the analyzes were performed (whether it has CE IVD certification, what controls were carried out, no exemplary results, no reference to results of other methods in these samples, e.g. IHC, no specificity, sensitivity, no information why these particular mutations would matter). I recommend publishing in a Turkish magazine.
Reviewer 3 Report
None